# Differences in Charge Distribution in *Leishmania tarentolae* Leishmanolysin Result in a Reduced Enzymatic Activity

**DOI:** 10.3390/ijms23147660

**Published:** 2022-07-11

**Authors:** Vítor Ennes-Vidal, Deborah Antunes, Ester Poláková, Vyacheslav Yurchenko, Simone S. C. Oliveira, Fabio Faria da Mota, Ana Carolina R. Guimaraes, Ernesto R. Caffarena, André L. S. Santos, Marta H. Branquinha, Claudia M. d’Avila-Levy

**Affiliations:** 1Laboratório de Estudos Integrados em Protozoologia, Instituto Oswaldo Cruz, Fundação Oswaldo Cruz (FIOCRUZ), Rio de Janeiro 21040-360, Brazil; claudia.davila@ioc.fiocruz.br; 2Laboratório de Genômica Funcional e Bioinformática, Instituto Oswaldo Cruz, Fundação Oswaldo Cruz (FIOCRUZ), Rio de Janeiro 21040-360, Brazil; deborah.santos@fiocruz.br (D.A.); carolg@ioc.fiocruz.br (A.C.R.G.); 3Life Science Research Centre, Faculty of Science, University of Ostrava, 70200 Ostrava, Czech Republic; estpolakova@gmail.com (E.P.); vyacheslav.yurchenko@osu.cz (V.Y.); 4Martsinovsky Institute of Medical Parasitology, Tropical and Vector Borne Diseases, Sechenov University, 119435 Moscow, Russia; 5Laboratório de Estudos Avançados de Microrganismos Emergentes e Resistentes, Instituto de Microbiologia Paulo de Góes, Universidade Federal do Rio de Janeiro (UFRJ), Rio de Janeiro 21941-902, Brazil; simonesantiagorj@yahoo.com.br (S.S.C.O.); andre@micro.ufrj.br (A.L.S.S.); mbranquinha@micro.ufrj.br (M.H.B.); 6Laboratório de Biologia Computacional e Sistemas, Instituto Oswaldo Cruz, Fundação Oswaldo Cruz (FIOCRUZ), Rio de Janeiro 21040-360, Brazil; fabio@ioc.fiocruz.br; 7Grupo de Biofísica Computacional e Modelagem Molecular, Programa de Computação Científica, Rio de Janeiro 21040-360, Brazil; ernesto.caffarena@gmail.com; 8Programa de Pós-Graduação em Bioquímica, Instituto de Química, Universidade Federal do Rio de Janeiro (UFRJ), Rio de Janeiro 21941-902, Brazil; 9Coleção de Protozários da Fundação Oswaldo Cruz (FIOCRUZ-COLPROT), Rio de Janeiro 21040-900, Brazil

**Keywords:** proteolytic activity, leishmaniasis, cloning, comparative modeling, molecular dynamics

## Abstract

*Leishmania tarentolae* is a non-pathogenic trypanosomatid isolated from lizards widely used for heterologous protein expression and extensively studied to understand the pathogenic mechanisms of leishmaniasis. The repertoire of leishmanolysin genes was reported to be expanded in *L. tarentolae* genome, but no proteolytic activity was detected. Here, we analyzed *L. tarentolae* leishmanolysin proteins from the genome to the structural levels and evaluated the enzymatic activity of the wild-type and overexpressing mutants of leishmanolysin. A total of 61 leishmanolysin sequences were retrieved from the *L. tarentolae* genome. Five of them were selected for phylogenetic analysis, and for three of them, we built 3D models based on the crystallographic structure of *L. major* ortholog. Molecular dynamics simulations of these models disclosed a less negative electrostatic potential compared to the template. Subsequently, *L. major LmjF.10.0460* and *L. tarentolae LtaP10.0650* leishmanolysins were cloned in a pLEXSY expression system into *L. tarentolae*. Proteins from the wild-type and the overexpressing parasites were submitted to enzymatic analysis. Our results revealed that *L. tarentolae* leishmanolysins harbor a weak enzymatic activity about three times less abundant than *L. major* leishmanolysin. Our findings strongly suggest that the less negative electrostatic potential of *L. tarentolae* leishmanolysin can be the reason for the reduced proteolytic activity detected in this parasite.

## 1. Introduction

Neglected diseases are a major public health problem in developing countries, including Brazil [1]. In the case of leishmaniasis, there are currently 12 million infected people worldwide, with an estimation of 700,000 to 1 million new cases occurring per year and 350 million people living at risk of infection [2]. At least 20 *Leishmania* species cause a wide spectrum of clinical manifestations, ranging from self-resolving skin lesions to life-threatening visceral diseases. In the pathogenic leishmania life cycle, the parasites are usually transmitted to vertebrates through the bite of a phlebotomine insect. In the mammalian host, *Leishmania* parasites have an obligatory intracellular form inside macrophages called amastigotes, but promastigotes in invertebrates change from a replicating procyclic to a non-replicating infective metacyclic form through a metacyclogenesis process [3,4]. The current therapy for leishmaniasis has serious side effects since it is limited to a handful of available drugs that suffer from administration difficulties, unacceptable toxicity, and in some cases, are ineffective because of the emergence of resistant strains [5].

*Leishmania* parasites are unicellular eukaryotes that belong to the Kinetoplastea class, family Trypanosomatidae [6]. Recent phylogenetic studies divided this protozoan into four distinct subgenera: *Leishmania*, *Mundinia*, *Sauroleishmania*, and *Viannia* [7]. *Leishmania* (*Sauroleishmania*) *tarentolae* was first isolated from the lizard *Tarentola mauritanica* in 1921, and it is the most widely studied *Sauroleishmania* [8]. It is presumed that *L. tarentolae* lives predominantly as promastigotes in the bloodstream or the lumen of the cloacae and intestine of the lizard host, and amastigote forms are rarely observed [9]. Although *L. tarentolae* promastigotes can enter human phagocytic cells and differentiate into amastigotes, no clear evidence for their efficient replication within macrophages was ever reported [10]. Due to its easy handling, fast growth in defined media, low maintenance cost, and lack of pathogenicity to humans, *L. tarentolae* has been extensively used as an experimental model for gene amplification studies [11], RNA editing [12], heterologous production of eukaryotic proteins [13], and development of vaccines [14]. However, its use to express virulence factors of pathogenic *Leishmania* species has been barely explored.

In this context, Raymond and co-workers [15] sequenced and assembled the *L. tarentolae* strain Parrot-TarII genome using next-generation high-throughput DNA sequencing technologies. The authors reported a high synteny of more than 90% gene content shared with other *Leishmania* species sequenced up to that time. Furthermore, some genes preferentially expressed in amastigotes, such as amastins and virulence factor A2, have been reported absent or present in fewer copies. In contrast, genes predominantly expressed in the promastigote form, such as the metallopeptidase leishmanolysin, were highly expanded in the *L. tarentolae* genome. Although the virtual translation of the leishmanolysin gene could be done, no enzymatic activity was recorded in cellular extracts of *L. tarentolae* by using the classical gelatin zymography assay [15].

Leishmanolysin, also known as a glycoprotein of 63 kDa (GP63), surface acid peptidase, promastigote surface peptidase (PSP), and major surface peptidase (MSP), is the major protein component of the surface of *Leishmania* promastigotes [16]. This protein is glycosylphosphatidylinositol (GPI)-anchored to the plasma membrane and has a zinc-dependent metallopeptidase activity, with a molecular mass ranging from 58 to 65 kDa. Due to the potentially relevant functions of leishmanolysin during the life cycle of leishmaniae and its therapeutic potential, since its discovery in the mid-1980s, this peptidase has been extensively investigated, and a myriad of functions facilitating parasite invasion, survival, and virulence have been described in parasite interactions with both invertebrate and vertebrate hosts [17,18,19]. In the genomic context, leishmanolysin exists as a multigene family sharing high sequence identities among different *Leishmania* species, which the number of genes ranges from only two in *L. (Leishmania) donovani* to a greatly expanded repertoire of 29 genes in *L. (Viannia) braziliensis* and 49 in *L. (Sauroleishmania) tarentolae* [15]. However, since no *L. tarentolae* proteolytic activity was reported by Raymond and co-workers, an intriguing question arises: is there an indirect correlation between gene expansion and enzymatic activity?

Fortunately, there is one available *L. major* leishmanolysin (*LmjF.10.0460*) three-dimensional structure (PDB ID: 1LML) deposited in the Protein Data Bank (PDB) [20]. The crystal structure revealed three domains (N-terminal, central region, and C-terminal) containing several conserved residues of around 60% identity compared to leishmanolysins from *T. brucei*, *L. major*, and *Crithidia* sp., as well as the zinc-coordination motif HEXXH [21]. The insertion of 62 amino acids in the defining motif HExxHxxGxxH differs, in *L. major*, from that commonly found in other metallopeptidases belonging to the metzincin class. Recently, to better understand the molecular basis of the expanded variety of chromosome 10 leishmanolysins of *L. braziliensis*, Sutter and co-workers (2017) [22] identified two levels of structural heterogeneity that affect the electrostatic properties of leishmanolysins and the geometry of their active sites. The authors assumed that the structural plasticity of *L. braziliensis* leishmanolysins might constitute a crucial evolutionary adaptation related to the ability of promastigotes to interact with a broad range of substrates [22]. In this work, to understand the lack of detectable proteolytic activity of the highly expanded leishmanolysin family in *L. tarentolae*, we analyzed the annotated leishmanolysin sequences and then created comparative 3D-models of *L. tarentolae* leishmanolysins to perform molecular dynamics simulations. Then, one *L. tarentolae* leishmanolysin and the one from *L. major* were overexpressed in an *L*. *tarentolae* expression system to improve our enzymatic knowledge of these molecules.

## 2. Results and Discussion

### 2.1. Unveiling the L. tarenrolae Leishmanolysin Annotated Sequences

Since the first isolation of the lizard Sauroleishmania parasite in 1921 [8], a great improvement in genome sequencing has taken place [10]. In this sense, Raymond and co-workers [15] used next-generation DNA sequencing technologies (NGS) to obtain a high-quality genome of *L. tarentolae*, revealing that more than 90% of the *L. tarentolae* genes are shared with all so far sequenced *Leishmania* species. However, some well-known limitations of this system were observed in the genome assembly, which results in a collapse of some repeat-rich regions and a tendency of repetitive expanded genes to be fragmented in the assembly, such as the leishmanolysin sequences [15]. Therefore, here we analyzed all the leishmanolysin genes annotated in the *L. tarentolae* Parrot-TarII genome available in the TriTrypDB.

We have retrieved a total of 61 sequences annotated as “Leishmanolysin” in the *L. tarentolae* reference genome (Appendix A), 12 more than previously reported due to the annotated contig sequences [15]. Interpro Scan predicted that these annotated proteins contain the metallopeptidase M8 domain in their sequences. However, 34 of these sequences carry less than two hundred amino acids, a sign of fragmented assembly. Furthermore, only eight of the remaining twenty-seven protein sequences have all amino acids recognized without any unknown residues (X), which appear when the annotated gene lacks an ascribed nucleotide. Therefore, considering the missing information in some sequences and focusing on the most conserved leishmanolysin sequences in our subsequent experimental methodologies, we first concentrated our analysis on these eight *L. tarentolae* leishmanolysin sequences, as reported in Table 1. Among them, one belonged to chromosome 31 (*LtaP31.2430*), one from chromosome 33 (*LtaP33.0300*), three from chromosome 10 (*LtaP10.0450*, *LtaP10.0480* and *LtaP10.0650*), and other three were contigs from unassigned chromosomes (*LtaPcontig00585-1*, *LtaPcontig00616-1,* and *LtaPcontig03442-1*) (Table 1). Chromosomes 31 and 33 share fewer identities with *L. major*
*LmjF.10.0460*, 35.6, and 58.4%, respectively. Such diversity is reasonably expected since the division of leishmanolysin sequences in different chromosomes probably happens in early steps during evolution [23]. The sequence identity for the other six sequences ranged from 63.2 to 71.2%, suggesting that the contigs could also be from *L. tarentolae* chromosome 10. Leishmanolysins on chromosome 10 constitute a set of gene arrays in all *Leishmania* species except *L. donovani* from the *Leishmania* subgenus, where it is absent. Interestingly, there are 42 leishmanolysins from chromosome 10 in *L. tarentolae* (Appendix A), suggesting that, as it occurs in species from the *Viannia* subgenus, it is highly possible that this leishmanolysin expansion took place after the separation of three subgenera and could be related to an adaptation mechanism of these species to interact with a broad range of reservoirs and vectors [22,23].

Considering the protein features from *L. tarentolae* leishmanolysin sequences predicted by bioinformatic tools, the molecular mass ranged from 24.0 to 66.1 kDa, and the disparity can be ascribed to the shorter or even lacking N-terminal domain in most of the sequences. Two leishmanolysins in chromosome 10 (*LtaP10.0450* and *LtaP10.0480*) presented one predicted transmembrane helix, while *LtaP31.2430* and *LtaP33.0300* displayed two helices. Moreover, four sequences (*LtaP10.0450*, *LtaP10.0650*, *LtaP31.2430*, and *LtaP33.0300*) had GPI anchor motifs predicted in their C-terminal domain. The presence of leishmanolysin molecules in the plasma membrane is critical to parasite adhesion and host cell interaction, as previously reported in several *Leishmania* species [18,19]. Only five sequences reported the presence of zinc-biding motifs (*LtaP10.0480*, *LtaP10.0650*, *LtaP31.2430*, *LtaPcontig00585-1*, and *LtaPcontig03442-1*), but the other sequences kept the conserved histidine and methionine residues, essential to the zinc-binding aligned with *LmjF.10.0460* at positions 334 and 345, respectively [21]. Finally, among the eight *L. tarentolae* leishmanolysins under analysis, the five most conserved sequences reported the presence of two associated domains by Interpro Scan, the Peptidase_M8 (PF01457) and LSHMANOLYSIN (PR00782).

To better understand the phylogenetic relationship of the leishmanolysins from *L. tarentolae* with the pathogenic *Leishmania* spp., we performed a phylogenetic reconstruction based on the Maximum Likelihood, Jones–Taylor–Thornton (JTT) model (Figure 1). Our results revealed that the five most conserved *L. tarentolae* sequences are closely related, supporting our hypothesis that the contig leishmanolysin sequences may have originated from chromosome 10. Moreover, *L. tarentolae* leishmanolysin protein sequences are phylogenetically more related to *L. major* than *L. braziliensis* and *L. martiniquensis* sequences. This was expected since molecular phylogenetic analysis suggests that *Sauroleishmania* is a sister group of the *Leishmania* subgenus [24]. In order to deepen the analyses, we selected the protein sequence most closely related to the *L. major LmjF.10.0460*, which has its three-dimensional structure revealed by crystallization and X-ray diffraction, to perform further studies.

### 2.2. Structural Heterogeneity on Charge Distributions in L. tarentolae Leishmanolysins

Comparative modeling resolved the *L. tarentolae* leishmanolysin three-dimensional shape to verify whether their differences affected the structural properties. The three-dimensional models were built based on the crystallographic structure of *L. major*
*LmjF.10.0460* leishmanolysin, displaying 72, 68 and 65% identity and 93, 90 and 96% coverage with *LtaP10.0650*, LtaP10.0480, and LtaPcontig00616-1, respectively (Appendix A). All models were refined, and their structural quality was assessed before and after the optimization procedure (Appendix A). Moreover, we obtained MolProbity scores ranging from 0.71 to 1.8 among the models, indicating no stereochemical clashes. In addition, all the obtained QMEAN values were higher than −4.23, which is the low-quality model threshold, making it appropriate for further study.

Like the *L. major* leishmanolysin three-dimensional structure, the *L. tarentolae* models are composed of three domains: N-terminal, central region, and C-terminal. These structures contain a zinc peptidase motif HExxHxxGxxH, such as Metzincin class zinc peptidases. However, an unexpected 62 amino acid insertion between the glycine and last histidine residue (Gly271–His334) of the metzincin motif represents most of the central domain [21]. The comparison between the template and the models revealed that the obtained structures displayed a root-mean-square deviation (RMSD) below 0.4 Å, meaning that the structures share a characteristic folding and are highly conserved (Figure 2). Leishmanolysins are rich in disulfide bonds. All structures presented nine conserved pairs, except for LtaPcontig00616-1, which presented eight pairs. The absence of the first disulfide bond pair is explained by the truncated N-terminal region, with 33 residues less than *LmjF.10.0460* (Figure 2).

The *L. major* template and *L. tarentolae* modeled 3D structures were submitted to molecular dynamics simulations to disclose the influence of pH on the net charge and stability of leishmanolysin. The pKa values of ionizable groups of *Leishmania* proteins were assessed for pH values 5.5 and 7.4. The RMSD values were computed to evaluate the influence of pH on leishmanolysin stability, resulting in an irrelevant difference below 0.3 Å (Appendix A). The template showed the highest variability when compared to the systems, oscillating at about 1.5 Å throughout the run. The models had an increasing mean of RMSD according to the lowest sequential identity with the template, displaying 2.3 ± 0.39 Å, 2.5 ± 0.47 Å, and 3.7 ± 0.75 Å for *LtaP10.0650*, LtaP10.0480, and LtaPcontig00616-1, respectively.

The pH influenced the net charge of *L. tarentolae* proteins, with more positive values for pH 5.5 and negative for pH 7.4 (Appendix A). The three active-site histidine residues that coordinate the zinc ion were assigned Nd neutral tautomers for all pH values. Non-catalytic histidine residues were positively charged at pH 5.5 and neutral at pH 7.4. (Appendix A). We also evaluated the electrostatic potential distribution on the surface of leishmanolysins. The *L. major* leishmanolysin showed an extensive region of negative electrostatic potential surrounding the active site (Figure 3). At less intensity, we find a similar pattern in *LtaP10.0650*, while LtaP10.0480 and LtaPcontig00616-1 showed a predominantly positive pattern of charge density. This difference between the electrostatic potential of *L. major* and *L. tarentolae* leshmanolysins could affect the protein orientation, as demonstrated for other proteins using experimental and theoretical approaches [25,26], as well as the impact on their affinities for substrates with distinct charge distributions [27]. Therefore, the proteolytic activity of *L. tarentolae* leishmanolysins, or at least the measurement protocols, might be affected by these less negative charge distributions. To confirm the charge distribution influence, *L. braziliensis* and *L. martiniquensis* leishmanolysin had their electrostatic potential analyzed. Similar to *L. major*, both leishmanolysins displayed a predominantly negative pattern of charge density (Appendix A).

Finally, to get a deeper understanding, we computed the residue solvent accessibility. Positively charged residues (lysine and arginine) presented higher SASA values (>1.5 nm/S2/N), with 6, 11, 11, and 13 residues for *LmjF.10.0460*, *LtaP10.0650*, LtaP10.0480, and LtaPcontig00616-1, respectively. We discovered that these residues led to the largest variance in charge density patterns compared to the electrostatic potential (Appendix A). As a result, changes in the electrostatic potential of amino acids occurred, particularly at the active site of leishmanolysins, changing their affinities for molecules.

### 2.3. L. tarentolae Leishmanolysin Has a Low Proteolytic Activity

Aiming to experimentally evaluate the molecular dynamics results of *L. tarentolae* leishmanolysins and to compare their proteolytic activity with *L. major* leishmanolysin (*LmjF.10.0460*), we cloned the most closely related leishmanolysin (*LtaP10.0650*) and the *L. major* one in pLEXSY expression systems to overexpress these peptidases (Appendix A). A hemagglutinin tag (HA tag) was added at the C-terminal of the cloned leishmanolysins, and two overexpressed clones were generated, as confirmed by the detection of the HA-tag (Appendix A). This expression system has been widely used for a broad range of biotechnological and biomedical applications due to the advantages of *L. tarentolae*. Its relatively easy and cost-effective cultivation, the mammalian-like post-translational protein modifications capable of producing substantial recombinant protein yields, as well as the potential to be extended to an industrial production scale are the main benefits that have attracted the attention of many researchers and make the *L. tarentolae* expression system such a good platform of heterologous protein production [28].

The metallopeptidase activity of wild-type *L. tarentolae* and *L. major* were first evaluated by gelatin-SDS-PAGE zymography. By incubating the soluble extracts in the gelatin gels for 72 h in phosphate buffer pH 5.5, we could observe a weak halo of degradation by the wild-type *L. tarentolae* extracts migrating in the 63 kDa range, which is remarkably close to the wild-type *L. major* polypeptides (Figure 4). These 63 kDa degradations were inhibited by the metallopeptidase inhibitor 1,10-phenanthroline and partially inhibited by the ion chelator EDTA, strongly suggesting the metallopeptidase activity (Figure 4). Although previous reports failed to detect proteolytic activity in *L. tarentolae*, this result is not surprising. Our research group has reported proteolytic activity in all monoxenic trypanosomatids and the phytomonads examined up to now, as well as all *Leishmania* spp. assessed so far [16]. Nevertheless, the activity is minimal compared to *L. major* (Figure 4) or to any other studied trypanosomatid. No difference was observed in gels incubated at pH 7.4 or even by the addition of DTT (data not shown). As expected, the metallopeptidase activity was higher in the two overexpressed *L. tarentolae* leishmanolysin strains and much higher in the two strains overexpressing *L. major* metallopeptidase, being incapable of complete inhibition by the metallopeptidase inhibitor at the concentration of 10 mM, probably due to the high amount of enzyme produced (Figure 4). This huge difference between the leishmanolysin activity from *L. tarentolae* and *L. major* might be attributable to the higher negative potential observed in the molecular dynamic simulations. Moreover, considering that several factors influence peptidase detection through gelatin-SDS-PAGE zymography [29], additional parameters were evaluated in our enzymatic assays, such as the buffer pH and the time of incubation, which could explain the differences between these results and those reported previously [15]. However, few details on the methodological approach were described, and it was assumed that the high sequence variability of the *L. tarentolae* leishmanolysin genes might have affected the peptidase activity. Accordingly, our research group previously observed no metallopeptidase activity incubating the gelatin gels with *L. tarentolae* extracts in an acidic buffer (pH 5.5) supplemented with 2 mM DTT for 24 h [30].

To better evaluate the differences between the proteolytic activity of *L. tarenrolae* and *L. major* leishmanolysins, we performed solution assays with the extracts of one clone of each overexpressed peptidase employing a specific matrix metallopeptidase substrate. After 40 min, the substrate was fully hydrolyzed by the soluble extract of the *L. major* leishmanolysin clone, and this proteolytic activity was almost completely inhibited in the presence of 1,10-phenanthroline, corroborating our previous findings in gelatin zymography (Figure 5). However, the overexpressing *L. tarentolae* leishmanolysin degraded about three times less substrate than the *L. major* enzyme. Furthermore, the presence of the serine peptidase inhibitor PMSF and the cysteine peptidase inhibitor E-64 had a slight effect on the activity of both leishmanolysin clones, while the ion chelators EDTA and EGTA partially affected the *L. major* peptidase (Table 2). Similar results were observed in the supernatant extracts from the spent culture medium (Appendix A). These results corroborate our hypothesis of reduced proteolytic activity of *L. tarentolae* leishmanolysin and raise a question about an indirect correlation between gene expansion and proteolytic activity. For instance, similarly to *L. tarentolae*, *T. cruzi* leishmanolysin suffer an extensive expansion to more than 420 predicted genes and pseudogenes, but the metallopeptidase activity of *T. cruzi* extracts is rather weak, only detected when the parasites have their soluble extracts obtained in the presence of proteolytic inhibitors from the other peptidase classes, such as PMSF, E-64, and the aspartyl inhibitor pepstatin A [31]. In contrast, pathogenic *Leishmania* spp. usually presents abundant detectable leishmanolysin activity and a lower number of leishmanolysin genes.

## 3. Methods and Materials

### 3.1. Sequence Analysis and Phylogenetic Reconstruction

Protein sequences of *Leishmania major*, *L. braziliensis*, *L. martiniquensis*, and *L. tarentolae* Parrot-TarII reference strain annotated as leishmanolysin (or GP63 or PSP or MSP) were retrieved from the TriTryp database (DB) [32]. These proteins were analyzed for InterPro families and domains databases by Interproscan v5.51-85.0 to confirm the presence of LSHMANOLYSIN (PR00782) and peptidase_M8 (PF01457) domains. The proteins containing both domains and more than 200 residues were aligned using MUSCLE v3.8.31. The predicted molecular mass of each sequence was calculated in http://www.bioinformatics.org/sms/prot_mw.html (accessed on 31 March 2016) and transmembrane helices were searched with http://www.cbs.dtu.dk/services/TMHMM (accessed on 31 March 2016). PROSITE (http://prosite.expasy.org, (accessed on 31 March 2016) was used to identify the zinc-binding motifs, and PredGPI (http://gpcr2.biocomp.unibo.it/cgi-bin/predictors/gpi/gpipe_1.4.cgi, (accessed on 22 March 2022) was used to predict the GPI-anchor. A phylogenetic reconstruction based on Maximum Likelihood, Jones-Taylor-Thornton (JTT) model with 1000 bootstrap replications, was performed using MEGA-CC v11 (NIH, Bethesda, MD, USA) and visualized in Dendroscope v3.5.7 (Tübingen, Germany).

### 3.2. Comparative Modeling and Molecular Dynamics Simulations

The structure of three *L. tarentolae* leishmanolysin proteins (*LtaP10.0650*, LtaP10.0480, and LtaPcontig00616-1), *L. braziliensis* LbrM.10.0600, and *L. martiniquensis* LMARLEM2494_000016500 were modeled. Initially, BLASTP was used to search the PDB database and select the 1.86 Å resolution crystal structure of *L. major* leishmanolysin (PDB ID: 1LML) [21] as a template. Then, the leishmanolysin sequences were aligned using the PSI-Coffee mode of the T-Coffee program, and five hundred homology models were created using the “standard auto model’’ routine of the Modeller v. 9.23 [33]. All modeled structures were optimized via the variable target function method (VTFM) with up to 300 iterations. Molecular dynamic (MD) optimization was conducted in the slow level mode, repeating the whole cycle twice to produce optimized conformations of the model. Finally, the modeled structures were filtered according to their discrete optimized protein energy (DOPE) score and refined using the locPREFMD server [34]. Original and optimized models were evaluated by the QMean server, while the ERRAT and Ramachandran plots were calculated on the SAVES server of UCLA-DOE Lab for stereochemistry analysis. Sequence alignment and three-dimensional structure figures were generated using ALINE and UCSF ChimeraX.

The three models of *L. tarentolae* and the template of *L. major* were simulated in the pH values of 5.5 and 7.4. MD simulations were carried out using AMBER 20.0; protein interactions were represented using amber ff14SB force field [35]. Protonation states of titratable residues were assigned using the PDB2PQR server. The disulfide bonds were specified. Electrostatic interactions were treated using the particle mesh Ewald (PME) algorithm with a cutoff of 12 Å. Each system was simulated under periodic boundary conditions in a triclinic box considering 12 Å from the outermost protein atoms in all cartesian directions. The simulation box was filled with TIP3P water molecules [36]. An appropriate number of Na^+^ and Cl^−^ counter ions were added to neutralize the systems. Subsequently, a two-step energy minimization procedure was performed: (i) 2000 steps (1000 steepest descent + 1000 conjugate-gradient) with all heavy atoms harmonically restrained with force constant of 5 kcal mol^−1^ Å^−2^; (ii) 5000 steps (2500 steepest descent + 2500 conjugate-gradient) without position restraints. All systems were further equilibrated during nine successive 500 ps equilibration simulations and simulated with no restraints at 300 K in the Gibbs ensemble with a 1 atm pressure using isotropic coupling. All chemical bonds containing hydrogen atoms were restricted using the SHAKE algorithm [37]. Three independent MD runs of 300 ns for each system were simulated using different initial velocities, achieving 7.2 μs simulation time. Trajectories were analyzed with GROMACS package tools. Root-mean-square deviation (RMSD) values were calculated separately for each system fitting their backbone atoms. Conformational clusterization for protein was performed using the GROMOS method with a cutoff of 2.0 Å, considering the backbone from the whole trajectory of each MD simulation. The central structure of the largest cluster was taken for the electrostatic potential analysis conducted with the Adaptive Poisson-Boltzmann Solver (APBS) program. Root-mean-square fluctuation (RMSF) and solvent accessible surface areas (SASA) were calculated for each system’s concatenated three independent MD simulations.

### 3.3. Leishmanolysin Cloning and Expression in L. tarentolae

*L. tarentolae* Parrot-TarII strain ATCC PRA-229 was maintained in M199 medium (Biowest, Nuaillé, France) supplemented with 2 μg/mL biopterin (Sigma-Aldrich, St. Louis, MI, USA), 2 μg/mL hemin (Sigma-Aldrich), 50 units/mL of penicillin, 50 μg/mL of streptomycin and 10% fetal bovine serum (FBS; Thermo Scientific, Waltham, MA, USA) at 26 °C with regular passages. For leishmanolysin overexpression, the pLEXSY system was used following the manufacturer’s instructions: genes for leishmanolysin from *L. tarentolae* (gene ID: *LtaP10.0650*) and *L. major* (*LmjF.10.0460*) were cloned into a vector pLEXSY-hyg2 (Jena Biosciences Inc., Jena, Germany) and tagged at the C-terminal with a hemagglutinin tag (HA3). The construct with *L. tarentolae* leishmanolysin (9 μg) and 7.7 μg of the *L. major* construct were taken and cut off with Swa I before transfection. Both PCR products were afterward purified with a PCR purification kit and used directly for transfection. Finally, 10^8^ cells were taken for transfection by electroporation (2 pulses, 25 μF, 1500 V, 10 s break between pulses) using BTX ECM 630 electroporator (Harvard Apparatus Inc., Holliston, MA, USA). Mutants were selected in an M199 medium supplemented with Hygromycin B (100 μg/mL) (Sigma-Aldrich).

As previously described, crude extracts of *L. tarentolae* mutants were analyzed by Western blotting to confirm the overexpression [31]. Briefly, proteins were separated in 10% SDS-PAGE, and the polypeptides were electrophoretically transferred to nitrocellulose membranes and blocked in 10% low-fat dried milk dissolved in PBS containing 2% Tween 20 (Sigma-Aldrich). The membranes were washed and incubated with an anti-HA antibody (Thermo Scientific) (1:1000 dilution) for 1 h. The secondary peroxidase-conjugated goat anti-rabbit immunoglobulin G (1:1500 dilution) (Thermo Scientific) was used, followed by chemiluminescence immunodetection. An anti-α-tubulin polyclonal antibody (Sigma-Aldrich) produced in rabbit (1:5000 dilution) was used as a loading control.

### 3.4. Parasite Cultivation for Protein Extraction

*L. tarentolae* Parrot-TarII strain wild-type strain (WT) and *L. major* (MHOM/SU/73/5-ASKH) promastigotes were routinely maintained in Schneider’s medium supplemented with 10% FBS at 26 °C. In addition, *L. tarentolae* mutants overexpressing leishmanolysin (*Lta* + *LtaP10.0650* and *Lta* + *LmjF.10.0460*) were maintained with 100 μg/mL hygromycin up to 20 passages, according to manufacturer’s instructions. To obtain the protein crude extracts, 10^9^ promastigotes from each strain were resuspended in 2% CHAPS (Sigma-Aldrich) diluted in 50 mM Tris-HCl, pH 7.2, and broken in a vortex by alternating 30 s shaking and 30 s cooling intervals [29]. The crude extract was centrifuged at 4 °C, and the soluble and insoluble fractions were separated by centrifugation (14,000× *g*). For the supernatant extraction, the spent culture medium was concentrated 50-fold in a 10,000-molecular-weight cutoff Amicon micropartition system (Amicon^®^, Sigma-Aldrich). The protein concentration was determined following the manufacturer’s instructions of Pierce™ BCA Protein Assay Kit (Thermo Scientific), using bovine serum albumin (BSA) as standard, and the samples were stored at −80 °C.

### 3.5. Enzymatic Activity Assays

The leishmanolysin activity of *L. tarentolae* mutants was assayed simultaneously by gel zymography and fluorimetry. First, approximately 40 µg of the soluble extracts from the supernatant were submitted to 12% polyacrylamide-SDS gel electrophoresis (SDS-PAGE) containing 0.1% gelatin from porcine skin (Sigma-Aldrich) in an ice-bath. Samples were diluted (*v*/*v*) in SDS-PAGE sample buffer 4× without reducing agent and loaded onto gels. Electrophoresis was carried out under constant current (25 mA). After running, the gels were washed 4 times in 2.5% Triton X-100 (Sigma-Aldrich) under agitation for 1 h at room temperature for SDS removal and enzyme renaturation. Subsequently, the proteolytic activity was determined by incubating the gels at 37 °C for 72 h in 50 mM phosphate buffer, pH 5.5 and pH 7.4, containing or not 2 mM dithiothreitol (DTT) (Sigma-Aldrich). To determine the peptidase class, the lysates were incubated with the inhibitors: (i) 1,10-phenanthroline (Phe, 10 mM), ethylene-bis(oxyethylenenitrilo)tetraacetic acid (EGTA, 10 mM) and ethylenediaminetetraacetic acid (EDTA, 10 mM) for metallopeptidases; (ii) Nα-tosyl-Lys chloromethyl ketone (TLCK, 100 µM) and phenylmethanesulfonyl fluoride (PMSF, 1 mM) for serine peptidases, (iii) and *trans*-epoxysuccinyl-L-leucylamido(4-guanidino)butane (E-64, 10 µM) for cysteine peptidases. The gels were stained with 0.2% Coomassie Brilliant Blue R-250 (Sigma-Aldrich) in methanol–acetic acid–water (40:10:50) and destained in the same solution without the dye. The gels were scanned and analyzed with Image Scanner III (GE HealthCare, Chicago, IL, USA). The molecular masses of the peptidases were estimated by comparison with the mobility of low molecular mass standards [29].

Alternatively, the enzymatic activity was measured continuously using the fluorogenic peptide substrate MCA-Pro-Cha-Gly-Nva-His-Ala-Dpa-NH2 (Sigma-Aldrich) at 37 °C in 0.1 M glycine-NaOH buffer pH 10.0 containing 1 mM CaCl_2_ [31]. The assays were performed in black round bottom 96-microwell plates in the Spectra Max Gemini spectrofluorometer (Molecular Devices, San Jose, CA, USA), with excitation wavelength at 328 nm and emission at 393 nm. The reaction started with the addition of 10 µM substrate to 10 µg of the parasite cellular extract, or the supernatant extract, in a total volume of 100 µL. The inhibition studies were performed by incubation with the following inhibitors: 10 mM Phe, 10 mM EDTA, 10 mM EGTA, 1 mM PMSF, or 10 µM E-64. In parallel, the addition of divalent ions ZnCl_2_, ZnSO_4_, CaCl_2,_ and MnCl_2_ at 5 mM was also evaluated. The assays were controlled for self-liberation of the fluorophore over the same time interval. All results were performed three times in triplicate and are expressed as mean ± standard deviation. In both zymography and in-solution assays, *L. tarentolae* WT and *L. major* cellular extracts were used as controls.

## 4. Conclusions

We performed a deep analysis of *L. tarentolae* leishmanolysins from the genome to the functional enzymatic studies. Our results unveil that the variations in the electrostatic potential between *L. major* and *L. tarentolae* leishmanolysins could be the main responsible for the reduced proteolytic activity of these peptidases. Nevertheless, the great expansion of *L. tarentolae* leishmanolysins genes and differences in the total charge distribution might modify the parasite–host interface, allowing evolutionary adaptation to different genus that serves as vector (*Sergentomya*) and the lizard host [23]. In addition, previous *L. major* results suggested that different substrate specificities displayed by these peptidases were a valuable characteristic to explain the importance of distinct leishmanolysin genes at *Leishmania* spp. surfaces [38]. Consequently, the evaluation of the rules played by *L. tarentolae* leishmanolysins in the interaction process with its hosts should be investigated.

In addition, we overexpressed one *L. major* and one *L. tarentolae* leishmanolysin in a pLEXSY system. The overexpressed parasites facilitated our understanding of the low metallopeptidase activity of *L. tarentolae* leishmanolysin. Trypanosomatids non-pathogenic to humans, such as *L. tarentolae*, could be used as important models for understanding the biological behavior, biochemistry, and molecular biology of pathogenic trypanosomatids [39]. Therefore, further studies on leishmanolysin in these species are warranted to better understand the function of these enzymes during the parasite life cycle and relate it to the lack of pathogenicity in vertebrates for these species.

## Figures and Tables

**Figure 1 ijms-23-07660-f001:**
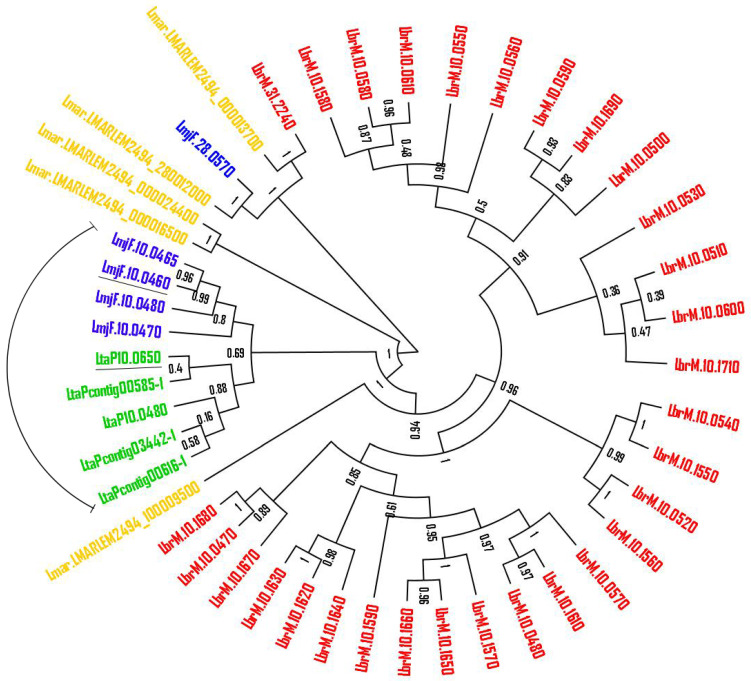
**Unrooted maximum likelihood phylogenetic tree using leishmanolysin protein sequences.***L. major*, *L. braziliensis*, *L. martiniquensis*, and *L. tarentolae* protein sequences annotated as “leishmanolysin” were obtained from TriTrypDB and were analyzed by Interproscan v5.16-55 to confirm the presence of LSHMANOLYSIN(PR00782) and Peptidase_M8(PF01457) domains. The sequences containing more than 200 residues were aligned and a phylogeny reconstruction based on Maximum Likelihood, Jones-Taylor-Thornton (JTT) was performed and visualized in Dendroscope v3.5.7. Numbers shown at branch nodes are bootstrap values based on 1000 replicates. The closed relation of *L. tarentolae* and *L. major* leishmanolysins are indicated, as well as the underlined leishmanolysins selected to be overexpressed.

**Figure 2 ijms-23-07660-f002:**
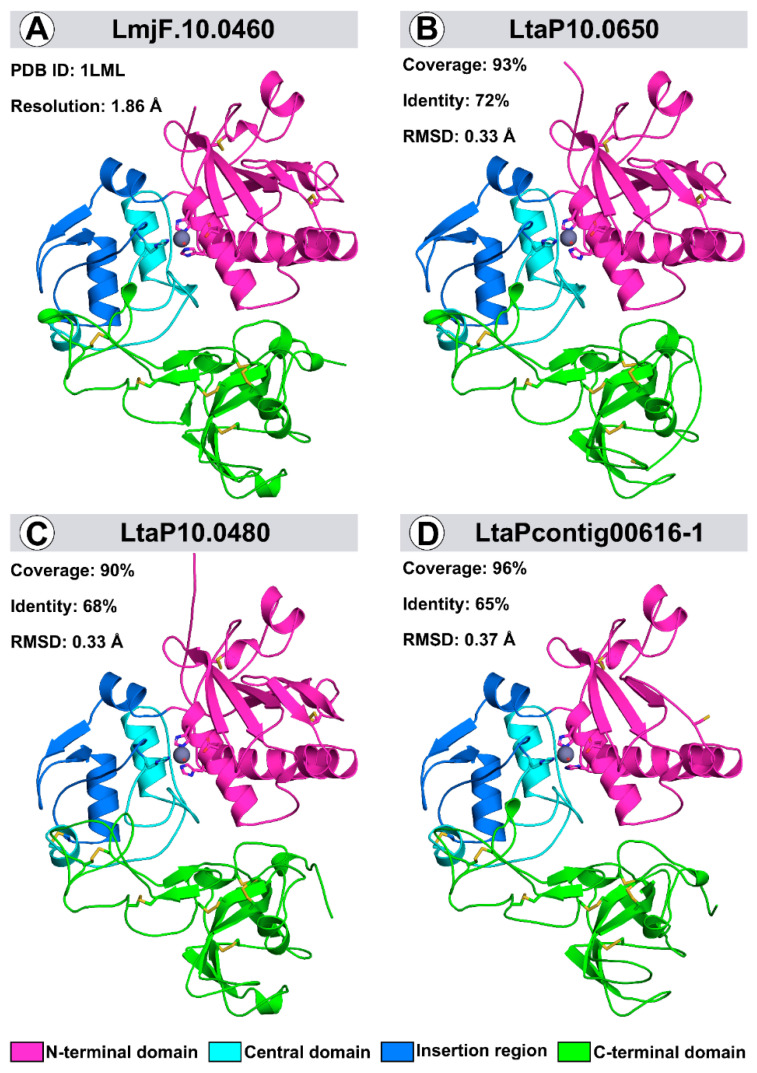
**Three-dimensional structures of *L. major* (PDB ID: 1LML) and *L. tarentolae* leishmanolysin 3D-models.** The PDB ID of the template and resolution are given in the upper left corner of the upper left quadrant, while the RMSD, coverage degree, and identity between models and the template appear in the upper left corner of the other three. The N-terminal, central region, and C-terminal domains are identified in pink, light blue, and green, respectively. The disulfide bonds are represented by sticks in yellow, the insertion region is identified in dark blue, and the zinc ion appears as a sphere in dark grey.

**Figure 3 ijms-23-07660-f003:**
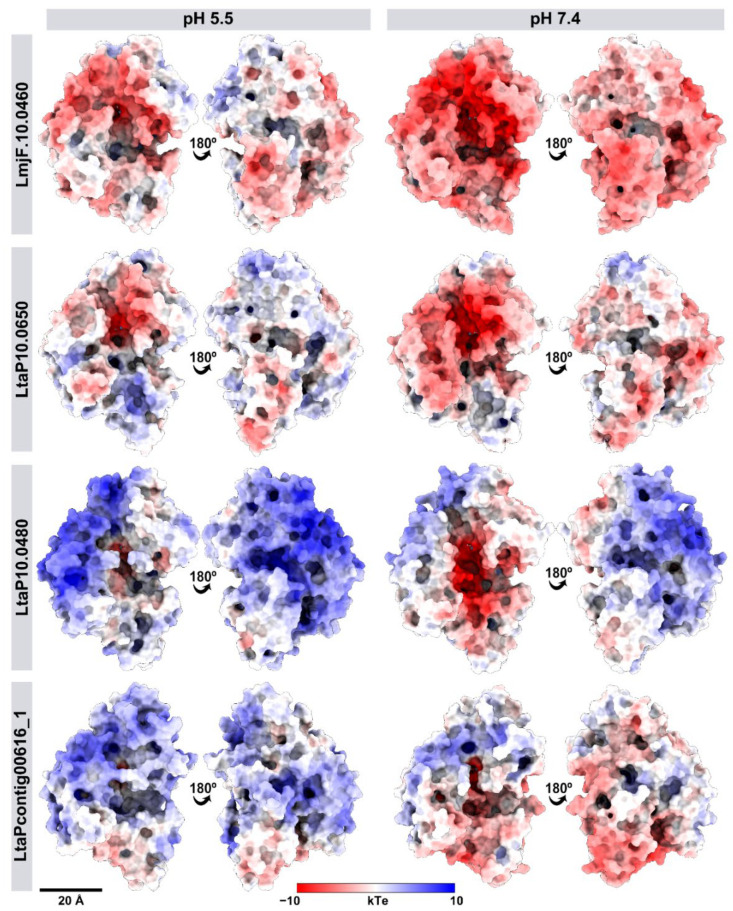
**Differences in the surface electrostatic potential of *L. tarentolae* and *L. major* leishmanolysins.** The comparative molecular dynamic analysis was performed using the crystal structure of *L. major*
*LmjF.10.0460* (PDBid: 1LML) and the modeled *L. tarentolae* LtaP10.0480, *LtaP10.0650*, and LtaPcontig00616_1 in pH 5.5 and 7.4. The molecular surface is colored according to electrostatic potential using the Chimera software, where red, white, and blue correspond to acidic, neutral, and basic potentials, respectively.

**Figure 4 ijms-23-07660-f004:**
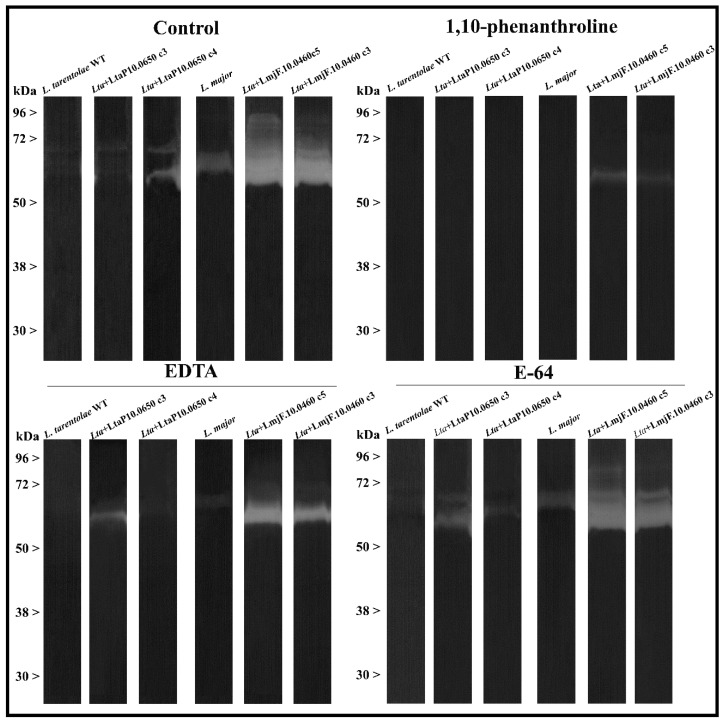
**Proteolytic activity of leishmanolysins from *L. tarentolae*, *L. major* and their mutants by Gelatin-SDS-PAGE zymography.** The soluble extracts of wild-type *L. tarentolae* (WT), *L. major,* and two isolates from each cloned leishmanolysins (*LmjF.10.0460* and *LtaP10.0650*) were submitted to Gelatin-SDS-PAGE electrophoresis. The gels were incubated in 50 mM phosphate buffer pH 5.5 for 72 h at 37 °C. To confirm the enzymatic class, zymograms were incubated in the absence (control) or presence of the ion chelators 1-10-phenanthroline and EDTA at 10 mM and E-64 at 10 µM to exclude cysteine peptidases activity around 63 kDa. The numbers on the left indicate apparent molecular masses of the active bands expressed in kDa.

**Figure 5 ijms-23-07660-f005:**
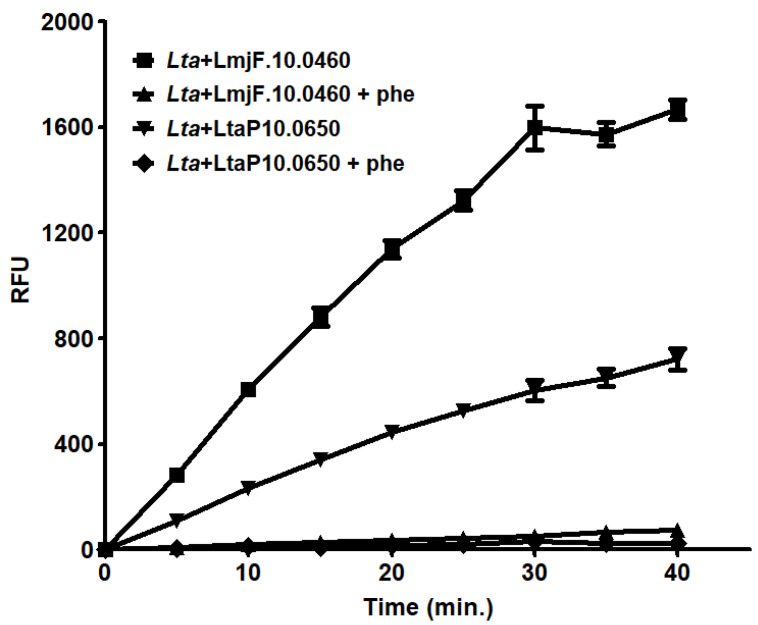
**In solution p****roteolytic activity of overexpressed *L. major*
*LmjF.10.0460* and *L. tarentolae LtaP10.0650* mutants.** The enzymatic activity was assessed by measuring the hydrolysis of the fluorogenic substrate MCA-Pro-Cha-Gly-Nva-His-Ala-Dpa-NH2 at 37 °C for 40 min. Activity of the cloned leishmanolysin from *L. tarentolae* (*Lta* + *LtaP10.0650*) and from *L. major* (*Lta* + *LmjF.10.0460*) were measured in 100 mM glycine-NaOH buffer pH 10.0 buffer the presence or absence of 1 μM1,10-phenantroline (*Lta* + *LtaP10.0650* + phe and *Lta* + *LmjF.10.0460* + phe). Results are expressed as relative fluorescence units (RFU) and the values represent the mean ± standard deviation of three independent experiments performed in triplicate.

**Table 1 ijms-23-07660-t001:** Leishmanolysin sequences from *Leishmania tarentolae*.

Sequence ID	Molecular Mass (kDa)	Domains	Transmembrane Helices	Zinc-Binding Motif	GPI-Anchored	Identity with *LmjF.10.0460*
*LtaP10.0450*	24.0	Peptidase_M8 (46-571 aa)	1 helix (287-308 aa)	*No hit*	YES	63.2%
*LtaP10.0480*	59.2	Peptidase_M8 (1-194 aa) LSHMANOLISYN (99-360 aa)	1 helix (9-28 aa)	1 hit (206-215 aa)	*No*	67.1%
*LtaP10.0650*	54.6	Peptidase_M8 (4-475 aa) LSHMANOLISYN (59-320 aa)	*No helix*	1 hit (166-175 aa)	YES	71.2%
*LtaP31.2430*	66.1	Peptidase_M8 (2-254 aa)	2 helices (13-30, 606-623 aa)	1 hit (268-277 aa)	YES	35.6%
*LtaP33.0300*	26.3	Peptidase_M8 (1-178 aa)	2 helices (67-91, 218-242 aa)	*No hit*	YES	58.4%
*LtaPcontig00585-1*	30.4	Peptidase_M8 (2-275 aa) LSHMANOLISYN (83-275 aa)	*No helix*	1 hit (190-199 aa)	*No*	68.7%
*LtaPcontig00616-1*	50.5	Peptidase_M8 (2-434 aa) LSHMANOLISYN (21-282 aa)	*No helix*	*No hit*	*No*	64.6%
*LtaPcontig03442-1*	42.1	Peptidase_M8 (3-381 aa) LSHMANOLISYN (59-321 aa)	*No helix*	1 hit (166-175 aa)	*No*	69.3%

The sequence ID of *L. tarentolae* leishmanolysin was obtained from TriTrypDB. The predicted molecular mass of each sequence was calculated in http://www.bioinformatics.org/sms/prot_mw.html, (accessed on 31 March 2016) and the presence of transmembrane helices was searched in http://www.cbs.dtu.dk/services/TMHMM, (accessed on 31 March 2016). Interproscan v5.51-85.0 was used to identify the presence of the classical leishmanolysin domains. PROSITE (http://prosite.expasy.org, (accessed on 31 March 2016) was used to identify the zinc-binding motifs, and PredGPI (http://gpcr2.biocomp.unibo.it/cgi-bin/predictors/gpi/gpipe_1.4.cgi, (accessed on 31 March 2016) predicted the GPI-anchor. The identity with *LmjF.10.0460* was calculated in CLUSTAL Omega (https://www.ebi.ac.uk/Tools/msa/clustalo/, (accessed on 22 March 2022).

**Table 2 ijms-23-07660-t002:** Effects of inhibitors on the proteolytic activity of overexpressed *Leishmania tarentolae* leishmanolysins.

Overexpressed Leishmanolysin	Inhibitor	Class of Inhibition or Concentration	Residual Activity (%)
*LtaP10.0650*	1,10-phenantroline	Metallo	2 ± 0.3
	EDTA	Metallo/Thiol	94 ± 1.2
	EGTA	Metallo/Thiol	101 ± 2.2
	E-64	Thiol	72 ± 3.1
	PMSF	Serine	101 ± 3.4
*LmjF.10.0460*	1,10-phenantroline	Metallo	3 ± 0.4
	EDTA	Metallo/Thiol	66 ± 1.6
	EGTA	Metallo/Thiol	66 ± 1.4
	E-64	Thiol	73 ± 2.9
	PMSF	Serine	73 ± 3.1

In solution assays were performed using the fluorogenic substrate MCA-Pro-Cha-Gly-Nva-His-Ala-Dpa-NH2 at 37 °C for 1 h. Activity of the cloned leishmanolysin from *L. tarentolae* (*LtaP10.0650*) and from *L. major* (*LmjF.10.0460*) in 100 mM glycine-NaOH buffer pH 10.0 buffer alone was taken as 100%. The effects of proteolytic inhibitors described below were measured according to the Materials and Methods.

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
