# Peer review of "Differences in Charge Distribution in Leishmania tarentolae Leishmanolysin Result in a Reduced Enzymatic Activity"

_ijms, 2022, doi:10.3390/ijms23147660_

Round 1
Reviewer 1 Report
The significance of the results is buried in details that do not add to the results. It is very hard to follow the authors interpretation of the data and in many cases there are alternative explanations for the results, although these are not mentioned. Overall I feel the data could have been analyzed better and the presentation lacks soundness.
Author Response
We appreciate your careful analysis of the data and we revised the MS accordingly. As mentioned above, the main objective of this study is quite simple since the weak GP63 activity from L. tarentolae is intriguing. The MS reported that the lack of more negative charged GP63 in L. tarentolae could be the main reason. Additionally, a more robust analysis including L. martiniquensis as a Mundinia subgenus representant was included in the reviewed version of the MS. The majority of the corrections suggested by the editor were accepted and so we hope our changes better clarify the MS interpretation. Please, find the modifications highlighted in the reviewed MS.

Reviewer 2 Report
Article written by Vítor Ennes-Vida et al. shows the differences in charge distribution in Leishmanolysin from Leishmania tarentolae and explains its reduced enzymatic activity. The authors conducted an exhaustive analysis of L. tarentolae leishmanolysins and concluded that differences in electrostatic potential between L. major and L. tarentolae leishmanolysins may be the reason for decreased activity. The article is very well written and interesting. I have only one question.
Are the leishmanolysins L. major and L. tarentolae present as dimers or as monomers and can this affect their activity?
Author Response
We are very grateful for the reviewer concerns. As previously mentioned, GP63 is synthesized in the endoplasmic reticulum, cleaved post-translationally, and has a GPI membrane anchor structure inserted at the last C-terminal 25 amino acid residues (Schlagenhauf et al., 1998). Approximately 75% of GP63 are located on the cell surface according to surface biotinylation, fluorescence microscopy, and immunoelectron microscopy (Weise et al., 2000; doi.org/10.1242/jcs.113.24.4587). Moreover, since the firsts secondary structure analysis performed by Bouvier and co-workers (1989; doi.org/10.1016/0166-6851(89)90155-2) GP63 has been reported as a soluble non-covalent homodimer.
Considering the proteolytic activity, most of the trypanosomatids have a GP63 that presents a robust activity easy to be detected by gelatin-SDS-PAGE zymography when incubated with the appropriate buffer. The fluorogenic substrate MCA-Pro-Cha-Gly-Nva-His-Ala-Dpa-NH2 (MMP13 – Sigma) has a great specificity to zinc-dependent endopeptidases that degrade matrix proteins, like GP63, being quickly hydrolyzed. A graduation student of our group standardized the better conditions to detect the enzymatic activity of L. tarentolae GP63 for around 4 years before we performed such in-depth proteolytic analysis. In addition, our research group have an extensive experience measuring GP63 activity in a variety of monoxenic trypanosomatids and phytomonads (d’Avila-Levy et al., 2014; Rebello et al., 2019).
Please, find the improvements in the MS revised version.

Round 2
Reviewer 1 Report
This is a greatly improved submission. The manuscript will be a valuable resource to studies in the area metalloproteinases in parasitic protists. The text has been improved considerably but there are still a few minor areas that will improve the clarity of the manuscript:
Results and Discussion section:
3.1 The opening paragraph needs revising:
“Since the first isolation of the lizard Sauroleishmania parasite in 1921 [8], a great improvement has been done [10].” I believe the authors are referring to advances in genome sequencing, if so it should be stated. “Since the first isolation of the lizard Sauroleishmania parasite in 1921 [8], a great improvement in genome sequencing has taken place”
4. Conclusions:
The opening statement of paragraph 1 - “In this study” - is redundant and can be deleted
The second paragraph - “In addition, here we overexpressed” here is not necessary and should be deleted.
The lettering above the lanes on Fig 4 is very small and hard to read on my screen.
Author Response
Dear reviewer, first of all, we would like to thank you for the careful analysis and criticism, which were used to improve the MS. A highlighted MS version containing the modified parts were upload to facilitate your analysis and a point-by-point answer to your concerns can be found attached.
